# Multiple Phase Stepping Generation in Alkali Metal Atoms: A Comparative Theoretical Study

Abu Mohamed Alhasan [1,2,*,†] , Abeer S. Altowyan [3], A. Y. Madkhli [4] and Salah Abdulrhmann [4,5,*]

1   Physics Department, Faculty of Science, Assiut University, Assiut 71516, Egypt
2   Bağlar Mahallesi, 31500 Ryhanlı, Hatay, Turkey
3   Department of Physics, College of Science, Princess Nourah bint Abdulrahman University, P.O. Box 84428, Riyadh 11671, Saudi Arabia; asaltowyan@pnu.edu.sa
4   Department of Physics, College of Science, Jazan University, P.O. Box 114, Jazan 45142, Saudi Arabia; amadkhli@jazanu.edu.sa
5   Department of Physics, Faculty of Science, Assiut University, Assiut 71516, Egypt
*   Correspondence: am.alhasan.sq@gmail.com (A.M.A.); sabdulrhmann@jazanu.edu.sa or abdulrhmanns@yahoo.com (S.A.)
†   Retired.

**Abstract:** We theoretically demonstrated optical phase switches in light storage-like experiments. Typical light storage (LS) and retrieval experiments consist of the probe field in the probe channel with writing and reading fields across the drive branch, as well as its recovery. The probe and first drive pulses as the standard electromagnetically induced transparency (EIT) effect of storing light are used in the proposed scheme for the atomic excitations. A train of probe pulses is used after a short storage period to induce Raman gain in the drive channel. The proposed scheme was applied to alkali-metal atoms such as $^{23}$Na, $^{87}$Rb, and $^{39}$K vapours. Spatiotemporal phase variations for generated drive pulses were found to shape in the form of discrete phase distributions. The proposed approach in the process of obtaining phase discrete distributions for different irradiation intensities was tested. For weak fields, the discrete distributions were distinct as a result of the differences in the upper hyperfine structure (hf) and the atomic relaxations. However, for moderate fields, the discrete phase distributions are smeared by the atomic relaxations.

**Keywords:** alkali-metal vapours; hyperfine structure; finite train propagation; phase-sensitive methods; uncertainty product; reduced Maxwell equations; density matrix equations

## 1. Introduction

Recently, much concern has been paid to applied quantum technologies (AQTs), with an emphasis on light storage (LS) and its restoring [1–5]. The LS effect is based on the quantum interference phenomenon [6–8]. Subject to conditions of electromagnetically induced transparency (EIT) [3,4], LS has been achieved experimentally through vapours [3,9–12] as well as in the solid state [13]. Notably, single photon storage and its restoring in $^{87}$Rb vapours have been shown by Buser et al. [1]. Korzeczek et al. have used a magnetic field to dominate the deviation of the reconditioned pulse [14]. Furthermore, the impact of the magnetic field on the radiation field's relative phase within an Λ system that interacted with two laser fields has been investigated [15]. In the absence of the magnetic field, the double-Λ system is not sensitive to the phase and is changed to the single Λ system. Once the magnetic field work, the double-Λ system becomes sensitive to the phase. Alternatively, Eilam et al. [16] considered a single Λ system with a probe, coupling, and an additional weak field between the lower hf levels. They found that the phase of the coupling field between the lower hf levels can control the amplification of the retrieved probe. In an inverted Y configuration, the observation of phase shifts over π was examined in a media with Rydberg excitons [17]. Alternatively, in a ladder scheme, Sinclair et al. [18] reported

the cross-phase modulation between dual optical pulses and measured the nonlinear phase written over the probe pulse. The phase-dependent EIT effect in the stationary regime for a diamond structure has been discussed by Korsunsky et al. [19]. Moreover, Radwell et al. [20] pointed out that standard EIT systems do not expose phase sensitivity. They demonstrated that, on applying a weak and transverse magnetic field that closes the EIT transitions, the phase-dependent transparency for the probe field is retained. In this article, we are mainly concerned with the phase that is created by the hf splitting structure in alkali-metal vapours and the phase produced from the pulses' reshaping during propagation. In a typical LS experiment, there is a probe pulse and coupling write pulse [3,4]. In our study, we have a probe and a writing driving field. The probe and the drive pulses might overlap in time or not. Additionally, in light retrieval typical experiments, there is the reading pulse, which is another drive field after some storage period. We have not considered such a reading pulse. Instead, we have considered a probe train in the probes channel. Throughout our analysis, we presume that the hf is determined, which is an appropriate presumption for cooled atoms [21].

## 2. Excitation and Atomic System Scheme

The atomic energy level scheme for alkali metal atoms such as $^{23}$Na, $^{87}$Rb, and $^{39}$K is depicted in Figure 1. The fine structure allowed transitions specified as $3\,^2S_{1/2}$—$3\,^2P_{1/2}$, $5\,^2S_{1/2}$—$5\,^2P_{1/2}$, and $4\,^2S_{1/2}$—$4\,^2P_{1/2}$ for $^{23}$Na, $^{87}$Rb, and $^{39}$K vapours, respectively. The kets $|1\rangle$ to $|4\rangle$ specifies the state of the hf levels, as depicted in Figure 1. $\Delta_p$ and $\Delta_r$ denote the detuning of fields from the transition $|1\rangle \leftrightarrow |3\rangle$ and $|2\rangle \leftrightarrow |3\rangle$.

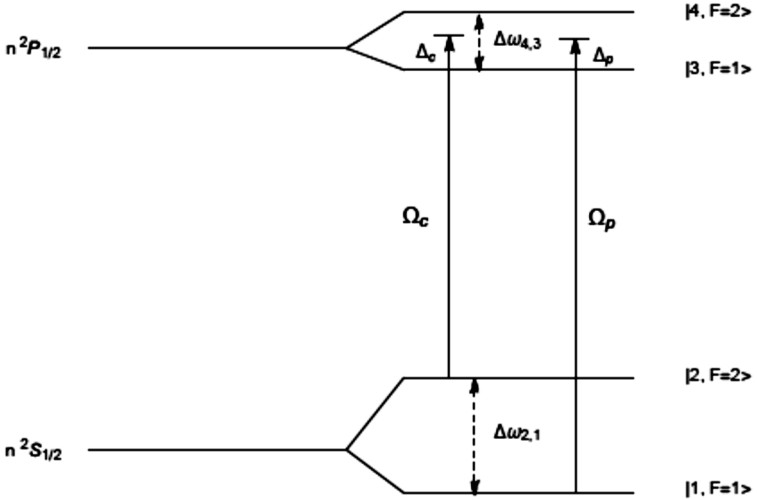

**Figure 1.** Optical transitions in the D$_1$ line hf-structure for $^{23}$Na, $^{87}$Rb, and $^{39}$K vapours.

The experimental data were taken from [22–25]. In detail, the experiment data for potassium were taken from [25] and those sodium from [23]. Furthermore, the experimental data for rubidium were found in [24], whereas the updated data were taken from [22].

The Liouville–von Neumann (LvN) equation [26,27],

$$-i\frac{\partial \rho_s(t)}{\partial t} = \hat{\mathcal{L}}_t \rho_s(t); \ \hbar = 1, \tag{1}$$

describes temporal development of the reduced density matrix (DM), $\rho_s(t)$, where $\hat{\mathcal{L}}_t$ stands for the Liouvillian super operator [28] in the Liouville space [29,30]. In the RDM description, we have adopted the formalism and notations in [28] and its extensions to the multilevel atoms with hf structures [31–33].

Our model variables are defined as follows: $(z, t)$ is the space and time coordinates in a laboratory frame, $\tau = \gamma(t - z/c)$ is dimensionless retarded time, $\gamma$ is the spontaneous decay rate of the atomic state P$_{1/2}$, $c$ is the speed of light, $\zeta = \alpha'(z + ct)$ is the dimensionless space

variable, $\alpha'$ is the absorption coefficient, $v$ is the atom-field, $\Omega = \sqrt{8}v$ is the Rbi frequency, and $\mathrm{v} = v/\gamma$ is the relative coupling of the atom-field.

The shapes for the probe and drive pulses for various vapours are shown in Figure 2. Along the boundary $\zeta = 0$, the pulses are formed as truncated-Gaussian pulses in time, which are characterized by different widths. We have used the driving pulse of one pulse, whereas the probing pulses formed a Gaussian train. Through $^{23}$Na vapour, we used pulses with envelopes that were not exactly matched in time, as the driving is narrower than the first probing pulse in the train. Let the pulse-width be denoted as $w$ and $\gamma$ is the spontaneous decay rate of the fine-structure transition. Consequently, the triples $\{\{\gamma_{\mathrm{Na}}, w_{\mathrm{Na}}\}, \{\gamma_{\mathrm{Rb}}, w_{\mathrm{Rb}}\}$, and $\{\gamma_{\mathrm{K}}, w_{\mathrm{K}}\}\}$ are obtained for the specific gases addressed in this study. The width compression in $^{87}$Rb and $^{39}$K gives rise to determined pulses through long times and near initial times. However, this is not true for the $^{23}$Na case, in particular for the first and second pulses. In the pulses' far-wings, the overlap is significant. During computations, we have kept the amplitude $\mathrm{v}_0$ the same, in agreement with $\frac{v_0^{Na}}{\gamma_{Na}} = \frac{v_0^{Rb}}{\gamma_{Rb}} = \frac{v_0^{K}}{\gamma_K}$. Thus, the pulses degenerate with respect to the amplitudes of the field. The degeneracy is removed as the intensity $I$ is expressed as $\left(\frac{\Omega}{\gamma}\right)^2 = \frac{I}{2I_s}$, where $I_s$ is the saturation intensity.

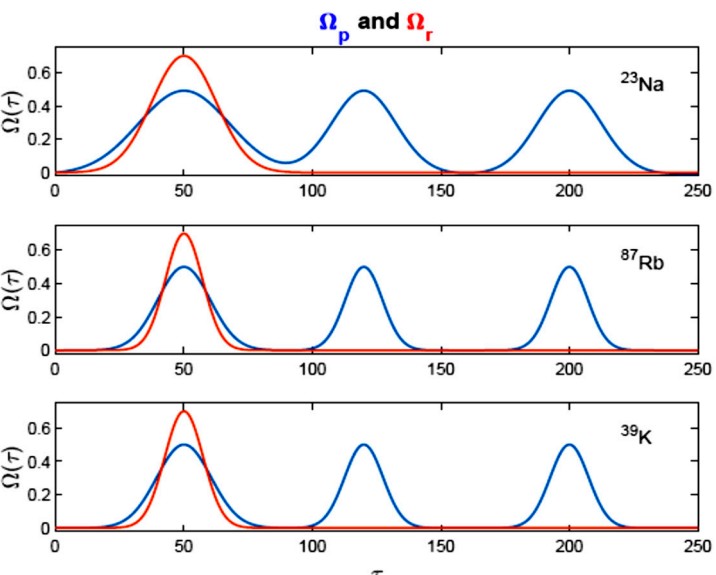

**Figure 2.** Pulse shapes for the probe, $\Omega_{\mathrm{p}}(\tau)$, and drive, $\Omega_{\mathrm{r}}(\tau)$, at the boundary $\zeta = 0$ for different vapours. Relative units are adopted for the time, Rabi frequencies, and detuning.

Temporal development is given by the LvN type equation and can be expressed in matrix form as

$$\frac{\partial \rho(t)}{\partial t} = L\big(\Delta\omega_{4,3}, \Delta\omega_{2,1}, \Delta_p, \Delta_r, v_p, v_r\big)\,\rho(t). \tag{2}$$

The behaviour of ultra-short pulses in the fine-structure regime subject to transition $3\,^2S_{1/2}$—$3\,^2P_{1/2}$ was considered [34]. The predominant equations are the coupled Maxwell–Bloch equations. The term Maxwell–Bloch is adopted as the slowly varying approximation (SVA) is applied to the Maxwell field. Moreover, the term Bloch equations are devoted to the reduced DM, incorporating a rotative wave approach (RWA). In the present case, with multilevel atoms and polychromatic field excitation, we shall separate the field part away from the atomic part. The optical transitions $|1\rangle \rightarrow |3\rangle$, $|4\rangle$ have one complex reduced Maxwell equation (RME) [35]. It is associated with the spatial evolution of the probe Maxwell field, $\Omega_{\mathrm{p}}(\tau)$. Similarly, we have a second RME, which is the space evolution of the driving field, $\Omega_{\mathrm{r}}(\tau)$. It corresponds to the transition $|2\rangle \rightarrow |3\rangle$, $|4\rangle$. Finally, the spatiotemporal propagation of the two field equations is coupled and abbreviated as the RM field equations for each specific alkali atom. Additionally, the description of space

evolution can be determined by three sets of reduced-field equations. As far as the SVA is concerned, the RMEs can be separated into

$$\frac{\partial v_p(z,t)}{\partial z} = \frac{\alpha'_p}{\sqrt{6}} \left[ \rho_{3,1}^{(10)}(z,t) - \sqrt{5}\rho_{4,1}^{(10)}(z,t) \right],$$

$$\frac{\partial v_r(z,t)}{\partial z} = \frac{\alpha'_r}{\sqrt{2}} \left[ \rho_{3,2}^{(10)}(z,t) - \rho_{4,2}^{(10)}(z,t) \right], \tag{3}$$

It is to be noted that the same RMEs and DMEs of the dressed-atom were also considered in connection to the LS effect and its retrieval in terms of entropy for the sodium atom [31]. The present study considers several alkali atoms such as sodium, rubidium, and potassium. The description presented here is adequate and we have considered alkali gases to have the same nuclear spin $I = 3/2$. The differences between the present and previous studies in [30] rely on the different mechanisms of field excitation to the atomic system.

Adopting dimensionless variables, the space evolution of the RMEs and the time evolution of the reduced DMEs can be written as follows:

$$\frac{\partial}{\partial \varsigma} v_p(\zeta,\tau) = \sqrt{6^{-1}} \left[ \rho_{3,1}^{(10)}(\varsigma,\tau) - \sqrt{5}\rho_{4,1}^{(10)}(\varsigma,\tau) \right],$$

$$\frac{\partial}{\partial \varsigma} v_r(\zeta,\tau) = \sqrt{2^{-1}} \left[ \rho_{3,2}^{(10)}(\varsigma,\tau) - \rho_{4,2}^{(10)}(\varsigma,\tau) \right],$$

$$\frac{\partial}{\partial \tau} \rho(\tau) = \frac{1}{\gamma} L \left( \Delta\omega_{4,3}, \Delta\omega_{2,1}, \Delta_p, \Delta_r, v_p, v_r \right) \rho(t). \tag{4}$$

Three RDM sets of equations corresponding to different vapours are taken into consideration. The atomic populations are kept in the first hf state initially, which is behind the so-called phaseonium medium [36]. For this medium, Clader and Eberly have obtained analytical forms for the ultrashort two-colour excitations in a single $\Lambda$ medium [37]. It is to be noted that the arrangements of probe and drive pulses are beyond the conventional EIT configuration. In addition, the theoretical and experimental results presented by Xu [15] and Eilam [16] demonstrated the phase sensitivity production by applying a magnetic field between the low-lying hf levels. In our approach, we attempted to achieve multilevel isochoric time-dependent sequences, without appealing to introducing the magnetic field. At the injection boundary $\zeta = 0$, we assume real-valued field envelopes, where both field and dipole phases are ignored.

## 3. Computational Results

### 3.1. Behaviour of the Amplitude and Energy for Probe and Driver Pulses Undergoing Propagation

We have adopted relative units. Rabi frequencies are expressed as $\Omega_p = 2\sqrt{2}v_p$ and $\Omega_r = 2\sqrt{2}v_r$ for probe and drive, respectively. In Figure 2, we have shown three time intervals for the interaction time of the atom with the radiation fields. An extension of the probe pulse is formed beyond the duration $T_1 = \{\tau \mid \tau \in [0, 85]\}$. The domain $T_2 = \{\tau \mid \tau \in [85, 170]\}$ specifies the second probe pulse in the train and is related to the first created pulse in the drive branch. The domain $T_2$ presents split pulses in the branch probes for $^{87}$Rb and $^{39}$K vapours. The last domain, $T_3 = \{\tau \mid \tau \in [170, 250]\}$, presents the domain for the second generated pulse.

We compute the relative energy of the probe pulses as

$$E_{p;T_\alpha}(\zeta) = \frac{\int_{T_\alpha} \Omega_{p;T_\alpha}(\zeta,\tau)d\tau}{\int_{T_\alpha} \Omega_{p;T_\alpha}(0,\tau)d\tau}, \quad \alpha = 1, 2, 3. \tag{5}$$

We present the gain in the drive channel according to

$$E_{r;T_\alpha}(\zeta) = \frac{\int_{T_\alpha} \Omega_{r;T_\alpha}(\zeta,\tau)d\tau}{\int_{T_1} \Omega_{r;T_1}(0,\tau)d\tau}, \ \alpha = 1,\ 2,\ 3. \tag{6}$$

Figures 3 and 4 expose the space dependence of the relative energies for both probe and drive pulses, where one is subtracted from the first drive pulse's energy to indicate the gain. In the representation $E_{1j}$, j = 1–3, the first suffix 1 indicates the lower (first) $\Lambda$ subsystem is concerned, as shown in Figure 1. Figure 3 exhibits the attenuation of probe pulses for different alkali vapours. Propagation in the probe channel is less attenuated in sodium vapours. Initially, the probe pulses have more energy than others. The attenuation in rubidium and potassium is similar, whereas depletion did not depend on whether or not pulses matched. Figure 4 displays amplification in the first drive pulse and gain in time sections $T_2$ and $T_3$, which is due to the generated Stokes field in the Raman transition. In alkali vapours, different gain mechanisms concerning hf structure can occur [38,39]. In the numerical results, we have focused our attention on the amplification of the generated Stokes field and the time dependence of its phase sequences, as shown in Figures 3 and 4. It is noted that, in our treatment, we have an incomplete LS experiment. This is because of the absence of the read pulse in the drive channel. Thus, the Stokes field generation is viewed as energy retrieval without invoking the traditional read pulse. Moreover, the phase sequence is associated with the restored Stokes field. The energy restored in the drive channel is written by the probe train. Our input pulses are not equivalent pulses, in the sense that it has not had the same energy and the same mean duration, initially. Therefore, it is difficult to compare the restoring efficiency of different alkali gases that can appear in alkali atoms.

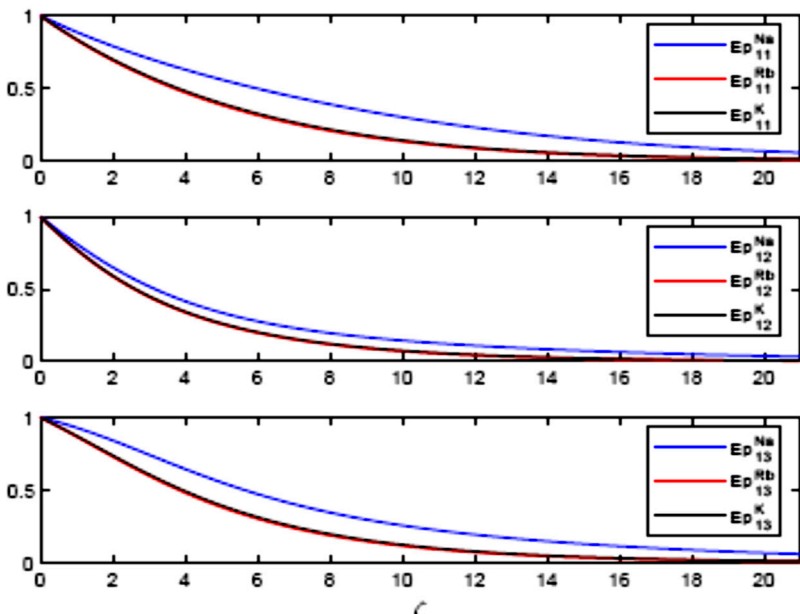

**Figure 3.** Attenuation of the relative energies for probe pulses in different alkali vapours. Energies $E_{11}$, $E_{12}$, and $E_{13}$ for probe and drive fields to correspond to time sections $\tau = [0, 85]$, $\tau = [85, 170]$, and $\tau = [170, 250]$, respectively.

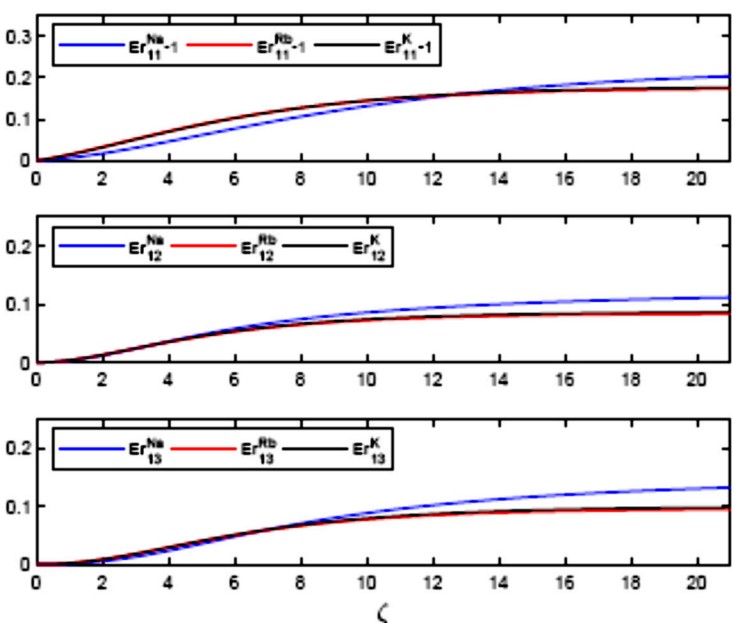

**Figure 4.** Amplification of the relative energies for drive pulses in different alkali vapours. Energies $E_{11}$, $E_{12}$, and $E_{13}$ for probe and drive fields to correspond to time sections $\tau = [0, 85]$, $\tau = [85, 170]$, and $\tau = [170, 250]$, respectively. We have subtracted one from the energy of the first drive component to show the gain.

### 3.2. Pulse Shaping and Uncertainty-Product Propagation

Figure 5 presents Gaussian shape profiles at moderate distances such as $\zeta = 21$. It is worth studying the uncertainty product propagation for created pulses within the drive branch associated with time intervals $T_2$ and $T_3$. The characteristic time of the pulse is defined as

$$\Delta \mathrm{T}(\zeta) = \left( \langle \tau \rangle^2_\Omega - \langle \tau \rangle^2_\Omega \right)^{1/2}. \tag{7}$$

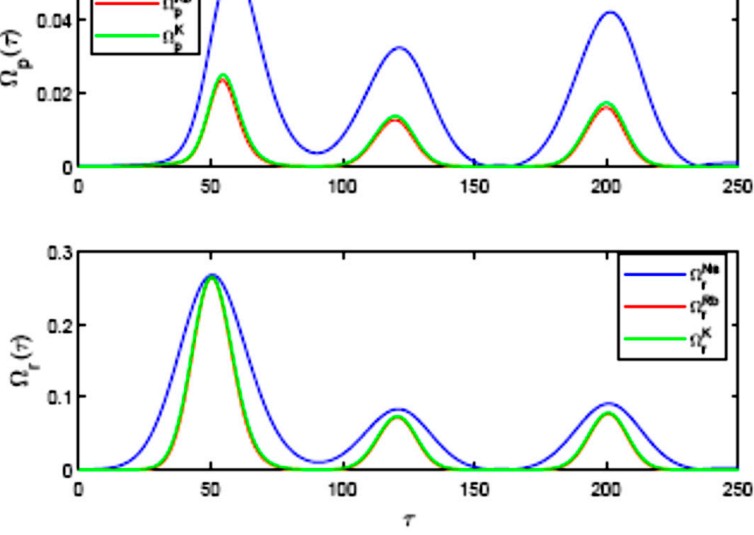

**Figure 5.** Pulse shaping and temporal behaviour of the field's amplitude absolute value in probe and the drive channels.

The dispersion in frequency gives

$$\Delta\omega(\zeta) = \left(\left\langle\omega^2\right\rangle_{\check{\Omega}} - \left\langle\omega\right\rangle_{\check{\Omega}}^2\right)^{1/2}. \tag{8}$$

The bracket in $\Delta T$ indicates the time average over the pulse profile. The bracket in $\Delta\omega$ indicates the frequency average over the Fourier transform of the pulse. The uncertainty product for Gaussian pulses becomes

$$\Delta\omega\Delta T \geq 1/2. \tag{9}$$

Figure 6 shows that generated pulses within the drive branch in both $^{87}$Rb and $^{39}$K vapours propagate with nearly a minimum uncertainty product. For sodium vapours, the pulses approach propagation with a minimum uncertainty product for a longer distance than that for rubidium and potassium. We have omitted the contribution to the uncertainty product for far times at the injection point $\zeta = 0$. There is no generation of pulses at such a point. The standard deviations for the time sections $T_2$ and $T_3$ are approximately zero at $\zeta = 0$. Figure 3 displays that the rate of energy loss is slower for sodium than for other vapours. Moreover, the gain is higher, as shown in Figure 4. Therefore, the uncertainty product for sodium takes longer distances to stabilize at the minimum product. For small distances, sodium pulses show a narrowing effect for the generated pulses in $T_2$ and $T_3$, while spreading occurs over long distances. Let us come to the discussion concerning the spatiotemporal phase. It is to be emphasized that both the probe and drive pulse are in resonance with their optical transitions. The phase generation is due to the upper hf splitting and the pulse shaping.

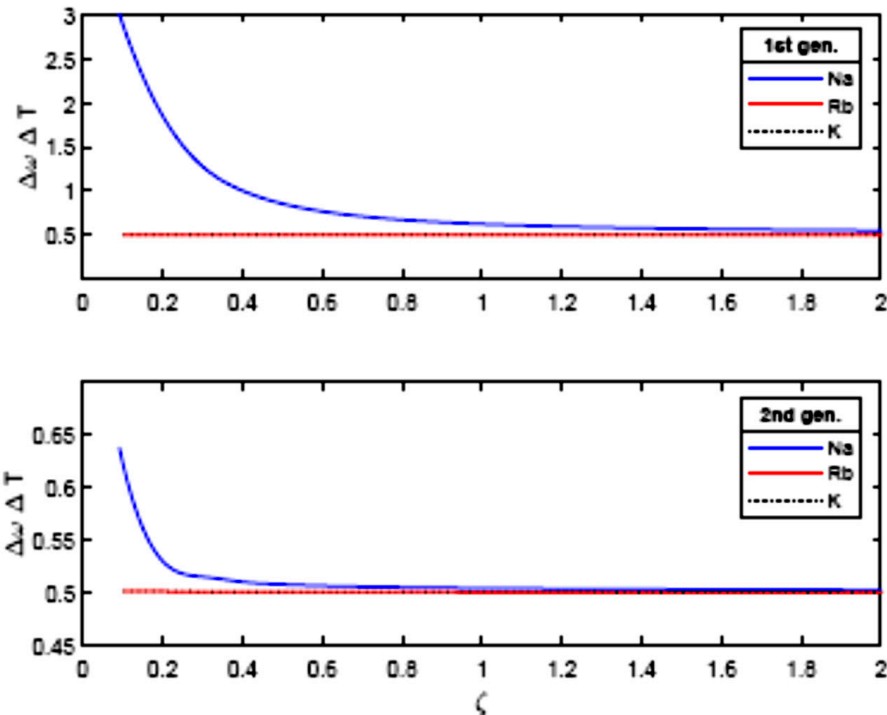

**Figure 6.** The drive pulses' uncertainty product propagation. The first (second) pulse generated is shown in the above (lower) panels, and corresponds to the time sections $T_2$ and $T_3$.

### 3.3. Phase Generation in a Perturbed Double $\Lambda$ Configuration

Figure 7 explores the time dependence of the phase patterns as generated through the drive channel counter to various distances through $^{23}$Na vapour. At the injection boundary, $\zeta = 0$, the hf-state $|3, F = 1\rangle$ is in resonance with the probe and drive. The results in phase trajectories are obtained upon simple standards, as the phases are generated by the

detuned-Λ, which acts as a disturbing subsystem. However, at the initial boundary, ζ = 0, the drive pulse shows phase changing from zero to π, at τ = 100, which is identical to the T₂ domain. The interference is dominant in the T₂ domain along the trailing edge of the first pulse and the front of the second pulse in the train. We notice that the phase is still zero at times in the classic EIT, T₁ part. We observed the drive pulses in the drive channel, such as T₂ and T₃, which propagate with different phase patterns through the medium for long time intervals, which makes it difficult to define a procedure for phase-pattern recognition. Intuitively, we may use the multilevel π-switches' regularities within the phase pattern as the distinguishable feature. We shall not investigate the width of the π-switches and the Euclidean time distance between multilevel switches. Hence, we have seven patterns, as shown in Figure 7.

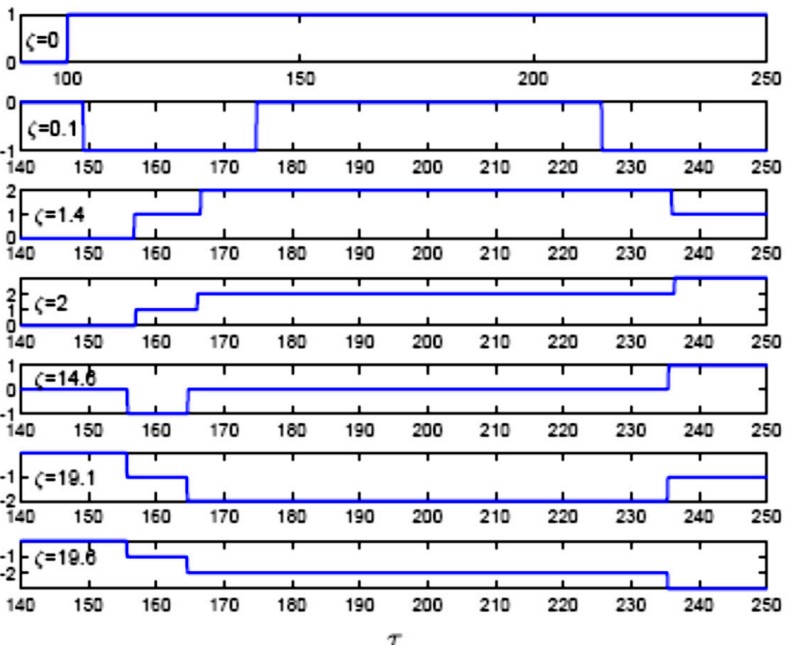

**Figure 7.** The trajectories of the temporal phase with the drive branch at various prolonged spaces in ²³Na vapour, where π units are used to express the phase.

The first sequence phase, $Q_1 = \{0, \pi\}$, is generated through interference among time sections, T₁ and T₂, at the injection point, ζ = 0. Phase patterns arise in different locations inside the gas medium. For small prolonged distances, such as ζ = 0.1, in relative units, we notice the phase pattern $Q_2 = \{0, -\pi, 0, -\pi\}$. The third pattern is devoted to the phase sequence $Q_3 = \{0, \pi, 2\pi, \pi\}$. The fourth pattern is considered as $Q_4 = \{0, \pi, 2\pi, 3\pi\}$ and generated at ζ = 2. Phase patterns are $Q_5 = \{0, -\pi, 0\pi, \pi\}$, $Q_6 = \{0, -\pi, -2\pi, -\pi\}$, and $Q_7 = \{0, -\pi, -2\pi, -3\pi\}$ at long distances. Concerning the interchange among phase creation and time intervals, interference between time sequences T₂ and T₃ produces phase patterns $Q_2$ to $Q_7$. Notably, the created ±π switch range is not the same for various sequences over the duration.

The generated pulses' relative phase about the corresponding probe pulses within the drive branches is given by

$$\Phi_{rp}(\zeta, \tau) = \frac{1}{\pi}\{\Phi_r(\zeta, \tau) - \Phi_p(\zeta, \tau)\} \tag{10}$$

Figure 8 shows the relative phase at the same space points as presented in Figure 7. We notice seven discrete phase distributions for alkali vapours, where, at the injection spot of pulses, $Q_1^{\{Na, Rb, K\}}$, the relative phase preserves the π value through the interaction interval T₁, which represents the period of the probe and drive first pulses. By following times, the switching of phase to 0π and π occurs again owing to interference becoming

considerable. The relative phases $Q_2$ to $Q_7$ are restitution between the phases because of no uniform pulse forming repairs and change in frequency of the upper hf-splitting at different prolonged distances in the medium. Through various trajectories of the phase, the $\pm\pi$ switches are split. Figure 8 shows the special pulse shaping impacts on the distribution of phase $Q_1$, which refers to deflections in phases. At far times, there is an increase in phase switches generated in $Q_2^{\{K,Rb\}}$, $Q_3^{\{K,Rb\}}$, and $Q_4^{\{K,Rb\}}$ for $^{87}$Rb and $^{39}$K vapours. Thus, we identify such variances among $^{87}$Rb and $^{39}$K as a group and $^{23}$Na. Over long distances, the relative phase switches in $^{87}$Rb and $^{39}$K become more stable at higher values than that for $^{23}$Na. The phase switch settles at $0\pi$ or $\pi$ for $^{23}$Na vapours, while for $^{39}$K and $^{87}$Rb, the separated distribution $Q_7^{\{K,Rb\}} = \{0\pi, \pi, \pi, 2\pi, 3\pi, 2\pi\}$ is found accordingly. Then, we notice two limits $0\pi$ or $\pi$ at $T_3$, for the relative phase in $^{23}$Na. Moreover, in $Q_7^{\{K,Rb\}}$, there are only $2\pi$ and $3\pi$ values. The rates of atomic spontaneous decay of $^{39}$K and $^{87}$Rb are close. The vapours of the $^{23}$Na have the highest value of the relaxation rate, and the upper hf splitting of $^{87}$Rb is enhanced 15 times compared with $^{39}$K. Therefore, our results depend on the rate of the spontaneous relaxation, and not on the upper hf splitting.

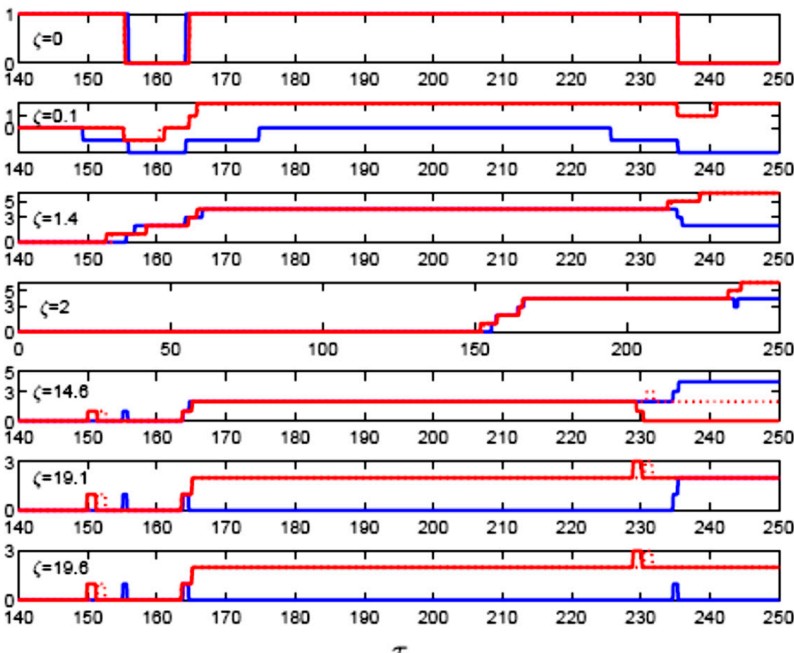

**Figure 8.** The drive pulses' relative phase with various prolonged spaces and expressed in $\pi$ units. The initial pulse profiles are depicted in Figure 2.

### 3.4. Discrete Phase Distributions Associated with Initially Weak Fields

Let us reveal the discrete phase distributions associated with initially weak fields. For such a case, the probe and drive pulses have the same shape as that in Figure 2. However, with amplitude reduction, initially at $\zeta = 0$, as $\Omega_p^{(max)} = 0.5/3$ and $\Omega_p^{(max)} = 0.7/3$. Figure 9 displays the relative phase in $\pi$ units. Different discrete phase distributions for alkali vapours are distinguished. The new behaviour was smeared for potassium and rubidium vapour in the moderate field case. The $\pi$ transitions are evident in Figure 9. Instead of plotting the shape profiles of the weak pulses, we shall write their effective fields at the desired distances as

$$\Omega_{eff}(\zeta) = \sqrt{8} \frac{\int |v(\zeta,\tau)|^2 d\tau}{\int |v(\zeta,\tau)| d\tau} \tag{11}$$

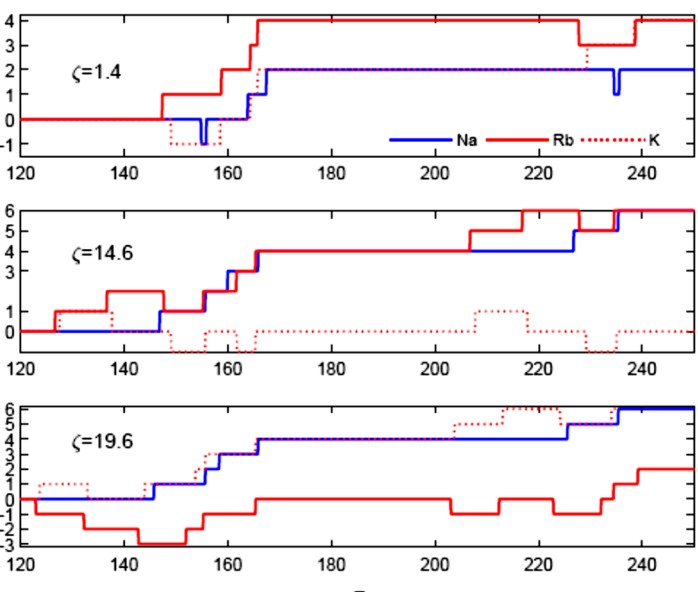

**Figure 9.** The drive pulses' relative phase with various prolonged spaces and expressed in $\pi$ units. (weak field's case).

For sodium and at $\zeta = 1.4$, the effective fields in relative units become $\left[\Omega^{p}_{eff}\ \Omega^{r}_{eff}\right] = [0.0875\ 0.1546]$. Moreover, at $\zeta = 19.6$, we have $\left[\Omega^{p}_{eff}\ \Omega^{r}_{eff}\right] = [0.0015\ 0.1452]$. The results for potassium and rubidium differ from that of sodium at the second-place decimal digit for the drive field and the fourth-place decimal digit for the probe field, which reflects phase sensitivity to small amplitude modulations. For the present case of initially weak fields, the asymptotic relative phases attain high values as $6\pi$, $4\pi$, and $2\pi$, indicating that the probe and drive pulses propagate with anti-phase.

*3.5. Enhancements of Higher-Rank DM Components*

In this study, we shall not encounter the redistribution of the scattered light into its multipolar expansion of the DM. However, it is pertinent to compare alkali vapours according to the enhancement in tensors of higher rank than orientation DM components. We briefly mention such efforts. The alignment-to-orientation conversion in rubidium vapour has been reported [40,41]. Recently, Urru and Spaldin have discussed the magnetic octupole-tensor together with the second-order magnetoelectric effect [42]. Alipieva et al. demonstrated the conversion between octupole and hexadecapole moments of the DM in rubidium vapours observed in fluorescence without stray magnetic fields [43]. Magnetooptical techniques can be used to selectively address individual high-rank multipoles, as discussed in [44].

Thoroughly, the discrete phase patterns and the generated pulses with minimum uncertainty products are formed with an enhancement of the high-order DM components. To establish that, we define the integrals

$$\Theta_{\alpha,T_{\beta}}(\zeta) = \int_{T_{\beta}} f_{\alpha}(\zeta,\tau)d\tau. \tag{12}$$

As an example, we take the function

$$f_{\alpha}(\zeta,\tau) = \{{}^{Na}\rho^{(2)}_{2,2}(\zeta,\tau),\ {}^{Na}\rho^{(4)}_{2,2}(\zeta,\tau),\ {}^{Rb}\rho^{(2)}_{2,2}(\zeta,\tau),\ {}^{Rb}\rho^{(4)}_{2,2}(\zeta,\tau),\ {}^{K}\rho^{(2)}_{2,2}(\zeta,\tau),\ {}^{K}\rho^{(4)}_{2,2}(\zeta,\tau)\} \tag{13}$$

We take the time periods as $T_1$, $T_2$, $T_3$, and $T = \cup T_{\alpha}$. The considerations of different $T_{\alpha}$ are significant to show the information transfer among different time sections. Specifically, we consider the whole interaction time T. For such a case, Figure 10 illustrates an

enhancement of the quadrupole and hexadecapole moments of the DM. These components represent losses. Therefore, we can conclude that the double $\Lambda$ system is inseparable owing to the enhancement of the high-order DM components. Figure 10 presents an enhancement of the third-order coherences describing a mutual relationship among lower-hf and upper-hf levels.

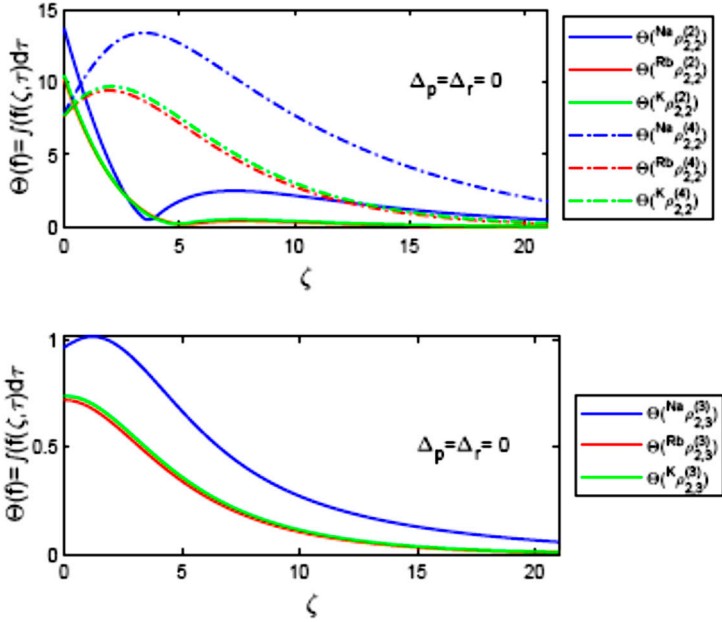

**Figure 10.** Space dependence of the time integral of quadrupole, octupole, and hexadecapole moments of the DM for different vapours. The initial pulse profiles are depicted in Figure 2.

## 4. Discussion

We have inspected the time dependence of the alkali-metal atoms' discrete phase distributions, which have the same nuclear spin ($I = 3/2$) and hf structure. We have proposed an incomplete procedure for the LS-like experiments, where the exciting pulses for relativity short times represent the probe and writing drive pulses, together with the addition of two pulses as a train within the branch probes without the read pulse necessity. Therefore, in the drive branches, Stokes pulses are created and propagated towards lower products of the uncertainties. This was influenced by atomic relaxations of all multipole orders, Figure 10. Initially, the system was sensitive to the phase, where the fields are real values. We have attained that, with management in the cross interaction of the pulses' wing sides, this leads to interference and phase changes. We have demonstrated that the upper hf splitting and the pulse shaping effects control discrete phase distributions. We have compared the discrete phase distributions for moderate and weak exciting fields and different alkali vapours. The created drive pulse phases and the relative phase of the drive with respect to the probe show a multi-probe show multilevel phase structure. This retrieved multilevel phase structure is formed through $\pm\pi$ phase switches.

For moderate drive and probe sequences, we have identified seven types of discrete phase patterns with $\pm\pi$-switches. These phase patterns during propagation are repeated in the same manner. We have defined a simple procedure for phase pattern recognition using the multilevel-$\pi$-switches' regularities within the phase pattern as the distinguishable feature. We have not considered the width of the $\pi$ switches and the Euclidean time distance between multilevel switches. Seven phase patterns, as depicted in Figure 7, are identified. In general, the multi-phase time sequences are generated beyond the EIT regime. For weak drive and probe sequences, the results of the multi-phase time sequences are more distinguishable than that of moderate fields and for different alkali-metal vapours. However, the splitting of the upper-hf state effect is dominant.

Consequently, we displayed the modulations of the digital phase in discrete sequences across different domain widths. The digital signals expand as constant phase rectangular $\pi$-pulses unless the step changes at the boundaries of the interval without ramping, i.e., the transitions of rectangular $\pi$-pulses are not smoothed. In relatively moderate fields, the jumps in phase approach $\pi$ and $0\pi$ for $^{23}$Na. Whereas the jumps approach $3\pi$ and $2\pi$ for $^{87}$Rb as well as $^{39}$K. Additionally, the levels of the phases approach $\{6\pi, 2\pi\}$ for relatively weak fields.

## 5. Conclusions

In this paper, we theoretically attempted to describe an LS-like experiment, in which there is a probe and driving write pulse, without appealing to the utility of a read pulse, as in the light retrieval experiment. Alternatively, we applied a probe train in the probe channel. Our computations for alkali atoms sharing the same nuclear spin $I = 3/2$ have shown discrete time distributions for the generated phases of the Stokes fields. Constant phase rectangular $\pi$-pulses excluding step changes at boundaries of the interval without ramping are a development of digital signals, which means rectangular stepping is not smoothed. Such phase distributions are obtained without appealing to magnetic field application between low-lying hf-states. We have tested our proposed approach in the process of obtaining phase discrete distributions for different irradiation intensities. For weak fields, the discrete distributions are distinct owing to the differences in the upper hf-states' structure and the atomic relations. However, for moderate fields, the discrete phase distributions are smeared and the only effect is due to the atomic relaxations. In atomic magnetometry, we hope that our computational approach can be used as a comparison tool.

**Author Contributions:** Conceptualization, A.M.A.; methodology, A.M.A., A.S.A., A.Y.M. and S.A.; software, A.M.A., A.S.A., A.Y.M. and S.A.; validation, A.M.A., A.S.A., A.Y.M. and S.A.; formal analysis, A.M.A., A.S.A., A.Y.M. and S.A.; investigation, A.M.A., A.S.A., A.Y.M. and S.A.; resources, A.M.A., A.S.A., A.Y.M. and S.A.; data curation, A.M.A., A.S.A., A.Y.M. and S.A.; writing—original draft preparation, A.M.A.; writing—review and editing, A.M.A., A.S.A., A.Y.M. and S.A.; visualization, A.M.A., A.S.A., A.Y.M. and S.A.; supervision, A.M.A.; project administration, A.M.A.; funding acquisition, A.M.A., A.S.A., A.Y.M. and S.A. All authors have read and agreed to the published version of the manuscript.

**Funding:** This research was funded by Princess Nourah bint Abdulrahman University Researchers Supporting Project number (PNURSP2023R16), Princess Nourah bint Abdulrahman University, Riyadh, Saudi Arabia.

**Acknowledgments:** Princess Nourah bint Abdulrahman University Researchers Supporting Project number (PNURSP2023R16), Princess Nourah bint Abdulrahman University, Riyadh, Saudi Arabia.

**Conflicts of Interest:** The authors declare no conflict of interest.

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
