# Peer review of "Multiple Phase Stepping Generation in Alkali Metal Atoms: A Comparative Theoretical Study"

_applsci, doi:10.3390/app13063670_

Round 1

Reviewer 1 Report

The manuscript can be improved taking the following comments into account:

1- Abstract : I think the abstract should not contain all these details. It should indicate the purpose of the study and the main findings.

2- The quality of some of the figures needs to be improved.

3- Figure 4: the position of this figure should be changed to be close to its referral in the text. Also the figure is not well discussed.

4- In line 248: why the width of the switch is not the same for different sequences? any justification? 

Author Response

Reply to Reviewer’s Comments

Reviewer 1#

We appreciate the contribution of the referee for the valuable comments. We have addressed all the REFEREE’S questions, and comments, and made the necessary changes. A list of changes made in the manuscript is underlined throughout the manuscript.

Comment# 1:

I think the abstract should not contain all these details. It should indicate the purpose of the study and the main findings.

Reply to comment# 1:

In the revised manuscript we have shortened the abstract indicating the purpose of the study and the main findings. A list of changes made in the manuscript is underlined throughout the manuscript.

Comment# 2:

The quality of some of the figures needs to be improved.

Reply to comment# 2:

In the revised manuscript, we have improved the quality of all figures. All changes made in the manuscript are underlined throughout the manuscript.

Comment# 3:

Figure 4: the position of this figure should be changed to be close to its referral in the text. Also the figure is not well discussed.

Reply to comment# 3:

In the revised manuscript we changed the position of Figure 4 to be close to its referral in the text.

Also, we added more discussions.

Comment# 4:

In line 248: why the width of the switch is not the same for different sequences? any justification?

Reply to comment# 4:

Phase production is due to the interference between the rising and falling edges of the neighborhood pulses resulting from the detuning of upper hyperfine splitting. Therefore the width of the generated switch is not the same for different sequences through propagation.

Reviewer 2 Report

Authors present computational study on the time-dependent phase distribution in a specific type of alkali atoms. I have several concerns related to this work.

1) Abstract - is misleading since no experiments were performed. So claim to demonstrate in first sentence is wrong. In general, abstract is a bit confusing and difficult to understand. I am missing what is connection of this study to applied sciences.

2) Introduction - first sentence claims that recently it is important but statement is again misleading. References are 20 years back and the phenomenon was of great interest like 10 more years back.  

3) Ref 13 & references therein - it is not common in scientific literature

4) Xu et al. then should be reference just after Xu et al. [15] etc.

5) page 2 line 65 sentence that data were taken from references ... is confusing. Please clearly state what data did you use from what reference

6) Eq. 1 - please add reference (e.g., Koo et al. doi: https://doi.org/10.1002/pssb.201100530)

7) Results and discussion should be essentially improved. It is just simple computing and not clear discussion related to application is given. Authors should more in deep discuss possible usage of their data.

Overall, I think manuscript requires significant revision before I can recommend it. 

Author Response

Reply to Reviewer’s Comments

Reviewer 2#

We appreciate the contribution of the referee for the valuable comments. We have addressed all the REFEREE’S questions, and comments, and made the necessary changes. A list of changes made in the manuscript is underlined throughout the manuscript.

Comments and Suggestions for Authors #:.

Authors present computational study on the time-dependent phase distribution in a specific type of alkali atoms. I have several concerns related to this work.

Reply to Comments and Suggestions for Authors #:

(a) The REFEREE points to "specific type of alkali atoms".  In this paper, we have considered the most important alkali atom gases that are frequently used for light storage effects and retrieval.

(b) The REFEREE points "specific".

This should be, for the sake of comparison between the selected gases. We have considered alkali gases with the same nuclear spin. It has the same set of density matrix equations.

Comment# 1a:

Abstract - is misleading since no experiments were performed.

Reply Comment# 1a:

The REFEREE pointed out that "no experiments were performed"

The experiments in light storage effects were mentioned in the manuscript in gases and solids. Please note that I am using "typical" meaning existing real experiments, not simulations.

Comment# 1b:

So claim to demonstrate in first sentence is wrong"

Reply Comment# 1b:

Is that a problem that I have used to demonstrate?

Comment# 1c:

In general, abstract is a bit confusing and difficult to understand.

Reply to comment# 1c:

In the revised manuscript we have shortened the abstract indicating the purpose of the study and the main findings. A list of changes made in the manuscript is underlined throughout the manuscript.

comment# 1d:.

I am missing what is connection of this study to applied sciences.

Reply to comment# 1d:

After our reply to the REFEREE comments, for sure the REFEREE can find such a connection.

Please note that we modified the manuscript title to be: “Multiple phase stepping generation in alkali metal atoms, a comparative theoretical study”. That is in order to fit with your opinion.

Comment# 2a:

Introduction - first sentence claims that recently it is important but statement is again misleading.

Reply to comment# 2a:

The abbreviation AQT for applied quantum technology has been used also by Stuttgart University.  http:/www.zaquant.unistuttgar.de

No misleading. We have considered light storage as one of these recent technologies.

Comment# 2b:

References are 20 years back

Reply to comment# 2b:

It depends; In fact, we have cited references, which are even in the press. A wide range of time is covered for theoretical and experimental works.

Comment# 2c:

the phenomenon was of great interest like 10 more years back. 

Reply to comment# 2c:

The REFEREE has not specified which phenomenon he is addressing. We have numerically, multilevel phase switches in alkali metal vapors. To the best of our knowledge; our current foundations have not been discussed earlier. The theoretical and experimental work of Elaim dealt with Pi phase production through the application of a magnetic field between the low-lying hyperfine levels. In the present communication, we obtained multilevel isochoric phase sequences. We have designated practical applications in quantum randomness generation.

Comment# 3:

Ref 13 & references therein - it is not common in scientific literature

Reply to comment# 3:

The sentence has been changed in the text.

Comment# 4:

Xu et al. then should be reference just after Xu et al. [15] etc.

Reply to comment# 4:

The statement has been changed in the text.

Comment# 5:

page 2 line 65 sentence that data were taken from references ... is confusing. Please clearly state what data did you use from what reference.

Reply to comment# 5:

From Ref. [25] we have taken the experimental data for potassium.  From Ref. [23] we have taken the experimental data for sodium. And from Ref. [24] we have taken the updated experimental data in Ref. [22] for rubidium.

Comment# 6:

Eq. 1 - please add reference (e.g., Koo et al. doi: https://doi.org/10.1002/pssb.201100530)

Reply to comment# 6:

Thank you for giving me the chance to read the article. It is of significant importance to solid-state devices. In fact, for such molecular media, the proposed article is far away from the current atomic media paper with a hyperfine structure. We addressed the time evolution of the density matrix equations as described by a Liouville-von Neumann  (Communicated with Prof. Jan Fiutak) equation for the dressed atom.  Therefore we have added one of the standard references to the von Neumann equation Ref. [26] and for educational purposes Ref. [27] in the manuscript.

26. von Neumann, John Wahrscheinlichkeitstheoretischer Aufbau der Quantenmechanik. Göttinger Nachrichten, 1927, 1, 245–272

27. Breuer H. P., and Petruccione, F. The theory of open quantum systems, 1st ed.; Oxford University Press: New York, USA, 2002; ch. 2, 110–117

Comment# 7a:

Results and discussion should be essentially improved.

Reply to comment# 7a:

It is done.

Comment# 7b:

It is just simple computing.

Reply to comment# 7b:

It is not simple. We have 28 equations for the reduced density matrix. Totally gives 28x3 of equations considered for different gases. It is not a numerically simple task to control in a self-consistent way the 28x3 density matrix equations in time evolution and 2x3 space evolutions for the Maxwell field equations. We only use parallel processing to make the computation faster. Simple computing is to consider a few equations for the density matrix and maybe without justifications.

Comment# 7c:

and not clear discussion related to application is given.

Reply to comment# 7c:

It has been pointed out in reply to comment# 2c and the following comment #7d.

Comment# 7d:

Authors should more in deep discuss possible usage of their data.

Reply to comment# 7d:.

Dear REFEREE, we have mentioned two experimental groups. The Eilam group in Ref [16] and the Xu et. al group in Ref [15].  Both groups utilized a magnetic field between the low-lying hyperfine levels in order to maintain phase sensitivity. Alternatively, in our approach, the discrete phase distributions are obtained without appealing to the application of a magnetic field between the low-lying hyperfine levels. After the storage period, we applied the probe train in the probe channel. It has been shown that the phase of the generated Raman fields in the drive channel exhibit multilevel discrete-time distributions. We hope that our proposed experiment based on our numerical demonstrations finds its potential application in comparison to magnetometer diagnostic of electromagnetic field phases.

Reviewer 3 Report

In this manuscript, the authors study phase and amplitude change of light pulse over space and time in alkali metal vapors. I cannot recommend publication of this manuscript in Applied Sciences for following reasons:

1. Figure 1 of this manuscript is a copy of figure 1 in Ref[29], only slightly changed labels.

2. Top two equations of Eq (4) of this manuscript is a copy of Eq(23) and (24) in Ref[29], again only slight changed variable name and subscript.

3. Authors didn't provide reference for so called "reduced-Maxwell field equations" in Eq(3)and Eq(4), I also can’t find a reliable reference. I put "reduced-Maxwell field equations" in quote marks did a exact match google search, the first page results are all papers of AM Alhasan. In other words, "reduced-Maxwell field equations" is created by AM Alhasan (the author of Ref[29]) , few people heard of it or use it, I can't tell whether "reduced-Maxwell field equations" is legit or not.

note:

Ref[29] Alhasan, AM. Entropy Associated with Information Storage and Its Retrieval. Entropy 2015, 17, 5920–5937 (https://www.mdpi.com/1099-4300/17/8/5920)

Thanks.

Author Response

Reply to Reviewer’s Comments

Reviewer 3#

We appreciate the contribution of the referee for the valuable comments. We have addressed all the REFEREE’S questions, and comments, and made the necessary changes. A list of changes made in the manuscript is underlined throughout the manuscript.

Comment# 1:

Figure 1 of this manuscript is a copy of figure 1 in Ref [29], only slightly changed labels.

Reply to comment# 1:

The REFEREE pointed out that figure 1 in the present paper is a copy of figure 1 in Ref [29]. This is true and should be. Both papers describe the same hyperfine structure of alkali metal atom associated with the fine structure D1 line and a nuclear spin I=3/2. In Ref [29] I have considered sodium vapor only and in the present article, I have considered sodium, rubidium, and potassium. These vapors differ in the upper and lower hyperfine splitting as well as the radiative and collisional relaxation rates. This has been pointed out in the figure caption of the present paper in figure 1. The associated fine structure transitions are also indicated in figure 1 of the present study.

Comment# 2:

Top two equations of Eq. (4) of this manuscript is a copy of Eq. (23) and (24) in Ref [29], again only slight changed variable name and subscript.

Reply to comment# 2:.

Again this is true and should be. Both papers describe two-color excitations, probe, and coupling fields.

The first equation, Eq. (4), is the same as Eq. (23). Both represent the reduced-Maxwell field equation for the probe transition. The second equation, Eq. (4) in the present paper is the same as Eq. (23) in Ref [29]. Both describe the reduced-Maxwell field equation for the coupling transition. The differences in the two papers rely on the difference in the initial conditions associated with the governing differential equations. In Ref [29] we have considered typical light storage experiment configurations. By which we mean that there are the probe and the coupling field (write) pulses. Following that a storage period. Finally, it comes out the read (coupling), another coupling pulse in the coupling transition.

In our present communication, we have described, for the first time, an incomplete light storage experiment. There is no coupling (read) pulse. Alternatively, we have considered a delayed probe train in the probe channel. Then we discussed the generated gain in Stoke's fields through the coupling (drive) channel. The distinction between the current and previous paper is outlined in the manuscript at the position (Line 165-176).

Comment# 3:

The authors didn't provide a reference for so-called "reduced-Maxwell field equations" in Eq(3) and Eq(4), I also can’t find a reliable reference. I put "reduced-Maxwell field equations" in quote marks and did an exact match Google search, the first page results are all papers of AM Alhasan. In other words, "reduced-Maxwell field equations" is created by AM Alhasan (the author of Ref [29]), but few people heard of it or use it, so I can't tell whether "reduced-Maxwell field equations" is legit or not.

Reply to comment# 3:

Thanks. It is true. In fact, the term reduced- Maxwell field equations has been addressed in many papers and accepted by many Reviewers both experimental and theoretical. The justification of the term is shown as follows.

The term reduced-Maxwell Bloch (RMB) equations for two-level atoms is adopted as the slowly varying approximation (SVA) is valid for the radiation Maxwell field and the rotating wave approximation is applied to the reduced density matrix.

The density matrix, which gives a good resemblance to the Density matrix equations, is reduced over the heat bath. So we have a reduced Maxwell equation and reduced density matrix equations for the two-level atom cases. The correspondence between Density matrix equations and the density matrix equations is obvious for two-level atoms. However, in my case as multilevel atoms, such correspondence may be ambiguous.

Therefore, I have not adopted the term Density matrix equations for multilevel atoms.   For multilevel atoms and polo chromatic field excitations, I have discussed the two terms: Maxwell equations and density matrix equations separately.   In the present study, I have two equations for the radiation fields and 6 equations for fields associated with different types of alkali atoms. In my approach I separated the description of fields away from the density matrix, I used the term Maxwell field equations to describe the radiation field and I used the term reduced density matrix equations for describing the state of the dressed atom by the fields.

I hope it is clear; however, I accept the REFEREE notion for such a case.

Round 2

Reviewer 2 Report

Authors have revised manuscript in accordance with suggestions. I have no further requirements.

Author Response

Reply to Reviewer’s Comments

Reviewer 2#

We appreciate the contribution of the referee for the valuable comments. We have addressed all the REFEREE’S questions, and comments, and made the necessary changes. A list of changes made in the manuscript is highlighted with yellow color throughout the manuscript.

Comment#:.

(x) Extensive editing of English language and style required

Reply Comment#

The manuscript has been checked by a colleague specialist in English.

Reviewer 3 Report

Authors answered my questions and made changes to the manuscript. I have one more comment, authors said "In fact, the term reduced-Maxwell field equations has been addressed in many papers and accepted by many Reviewers both experimental and theoretical." I hope authors can add some reference for "reduced-Maxwell field equations", so readers can get a better understanding. 

Author Response

Reply to Reviewer’s Comments

Reviewer 3#

We appreciate the contribution of the referee for the valuable comments. We have addressed all the REFEREE’S questions, and comments, and made the necessary changes. A list of changes made in the manuscript is highlighted with yellow color throughout the manuscript.

Comment# 1:.

authors said "In fact, the term reduced-Maxwell field equations has been addressed in many papers and accepted by many Reviewers both experimental and theoretical." I hope authors can add some reference for "reduced-Maxwell field equations", so readers can get a better understanding. 

Reply to comment# 1:.

We have abandoned using the so-called reduced Maxwell field equations. Alternatively, we used reduced Maxwell equations. The last term is widely used in different connections. The Referee may search using his preferred Google search engine to find its implementations and consequences.

In fact, Dr. Alhasan used a different term to in focuses the scientific reader to concentrate on his formalism for the irreducible set of the density matrix for the specific D1 transition and nuclear spin I=3/2, as he found it.  This is beyond density matrix foundations that are not irreducible. And do not take into account the multiplicity of the hyperfine levels. Or take the equality of the relaxation rates for the excited states.

The referee asked about some references.

Please see Ref. [35] in the manuscript. A. I. Maimistov book for the availability of the term reduced Maxwell equations and its single polarization component for each Maxwell field as well as its associated Bloch equations.

35. A. I. Maimistov, A. M. Basharov, Nonlinear Optical Waves, 1st ed.; Kluwer Academic Publishers: Springer Science+Business Media Dordrecht, Germany,1999; pp. xi, 29, 30, 33, 229 and 293.